# The Method of Determining Layer in Bottom Drainage Roadway Taking Account of the Influence of Drilling Angle on Gas Extraction Effect

**Yuliang Yang [1,2], Penghua Han [1,*], Zhining Zhao [1] and Wei Chen [1]**

[1] School of Energy & Mining Engineering, China University of Mining and Technology (Beijing), Beijing 100083, China; yyldtdx@163.com (Y.Y.); zzn23364540262022@163.com (Z.Z.); wwck2340@126.com (W.C.)

[2] School of Coal Engineering, Shanxi Datong University, Datong 037003, China

* Correspondence: 18800149959@163.com

**Abstract:** The pre-drainage of coalbed methane through boreholes in the bottom drainage roadway (BDR) is the key measure to prevent and control coal and gas outburst. Different arrangement layers in the BDR will make a difference in the range of drilling angle and affect the gas extraction effect. In this paper, the mathematical model of the rock loose circle area around elliptical drilling was constructed. Meanwhile, the fluid–solid coupling model is constructed by using COMSOL software, the dynamic response of coal permeability and volumetric strain with gas pressure and the Klinkenberg effect of gas are considered, and the effect of the change of the elliptical drilling angle on the pressure relief effect of the coal seam is studied. The results showed that the distance between the layer in the BDR and the pre-drainage coal seam would decrease, and the effective extraction length at the same point of gas extraction in the coal seam increases. The area of the rock loose circle and permeability around the drilling decayed negatively and exponentially with the increase in drilling angle. As the drilling angle decreased, the stress in the major axis of the ellipse at the drilling cross-section increased, so the drilling was prone to collapse, and the gas extraction was hindered. Finally, an optimal method of determining the layer in the BDR under the coupling effect of multiple factors was established by combining the measured ground stress. Through field measurement, the drilling extraction rate of the optimized scheme is 60% higher than that of the original scheme.

**Keywords:** drilling angle; layer; area of rock loose circle; permeability; bottom drainage roadway

## 1. Introduction

The pre-drainage of gas in the coal seam through the boreholes in the bottom drainage roadway (BDR) is one of the major measures to prevent and control coal and gas outburst [1–3]. As an important parameter that affects the gas extraction effect, the layer in the BDR needs to be set reasonably during the design and construction processes. Different drilling angles will make different exposure areas of coal and pressure relief degree of the coal seal segment, which affects the gas extraction effect of drilling greatly. Therefore, the control effect of the layer in the BDR on the drilling angle will indirectly affect the gas extraction effect. The method of determining layer in the BDR, taking account of the influence of drilling angle on the gas extraction effect, is of instructive significance to the arrangement of the boreholes for gas extraction in the BDR.

Currently, scholars mainly study the influence of drilling angles on gas extraction through theoretical analysis and numerical simulation [4,5]. Most studies focus on the relationship between extraction radius with diameter, inclination, pre-drainage time and negative pressure of the borehole stating that with the increase in drilling angle, gas extraction radius tends to decrease and then increase [6,7]. In addition, gas extraction flow is very sensitive to the effects of ground stress and mining stress. Yin et al. [8] found that

gas extraction flow decayed negatively and exponentially with the increase of the coal seam bearing stress. Dong et al. [9] studied the effect of drilling angle for pre-drainage on gas extraction effect based on the distribution of stress in the wall rock. Their study showed that the reduction of drilling angle was conductive to gas release in coal. Except for stress, the gas extraction amount of the coal seam was also determined by original gas pressure, extraction negative pressure and permeability of coal [10–12]. In terms of layer determination in the BDR, Liu et al. [13] proposed to arrange the BDR along the strike of the working face and use ultra-high pressure hydraulic cutting and permeability enhancement technology for pre-drainage in the coal roadway. Zhang et al. [14] studied the effect of seepage direction on a permeability stress test and the axial and radial permeability testing of coal under cyclic loading and unloading. Zhao et al. [15] studied the location of the roadway from the perspectives of wall rock stability in the roadway and prevention of coal and gas outburst caused by the BDR through the undulating coal seam. Nan et al. [16] studied the stress distribution and roadway deformation mechanism in the BDR based on FLAC3D numerical simulation and determined the layer in the BDR according to economic factors.

On the basis of the determination of the layer of the BDR, some scholars have studied the influence of the drilling angle on the attenuation law of gas drainage. Other scholars use the surrounding rock geological condition and stability analysis to determine the layer of the BDR. In this paper, taking the 8406 longwall face of Yangmei Coal Mine as the engineering background, based on the end point of the borehole and the range of gas drainage, the influence of the drilling angle on the area of the loose circle around the elliptical drilling is analyzed. The fluid–solid coupling model is constructed by using COMSOL software. The model considers the dynamic response of coal permeability and volumetric strain with gas pressure and the Klinkenberg effect of gas, and the effect of the change of elliptical drilling angle on the pressure relief effect of the coal seam is studied. Finally, the reasonable position of the BDR is determined by measuring the in-situ stress. The calculation results are verified with the regulations on prevention and control of coal and gas outburst so as to achieve the optimal drainage effect and reduce the engineering quantity and construction cost.

## 2. Influence of Drilling Angle on Area of Rock Loose Circle around Drilling

### 2.1. Theoretical Analysis

The drilling formed in coal seams will change the stress state around the drilling for form crushing zone, plastic zone, elastic zone and original rock stress zone around the drilling. The crushing zone and plastic zone are collectively called rock loose circle. In the rock loose circle, coal is destructed, fractures are developed, which is conductive to gas seepage in the coal seam. Therefore, the analysis of varying patterns of rock loose circle around the the borehole with a drilling angle is of important significance for the study of the gas extraction effect.

The arrangement of the borehole in the BDR of the coal seam is shown in Figure 1a. As shown in Figure 1b, the projections of three sets of drilling with different inclination angles change in the horizontal cross-section, and the projections of drilling in the vertical cross-section are circular, while the projections of drilling intersecting with the cross-section are elliptical and change with the angle. Let the diameter of drilling be a. If the diameter of drilling stays the same, the minor semi-axis of the ellipse does not change in the projection of inclination angles, and its value is a. The major semi-axis b changes with inclination angles, as shown in Figure 2.

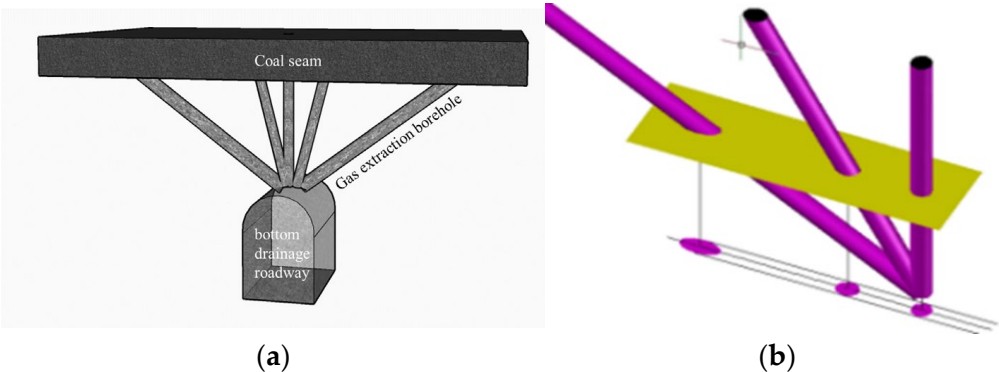

**Figure 1.** Arrangement with different drilling angle. (**a**) Arrangement of borehole in BDR of the coal seam, (**b**) Projection of horizontal section of borehole.

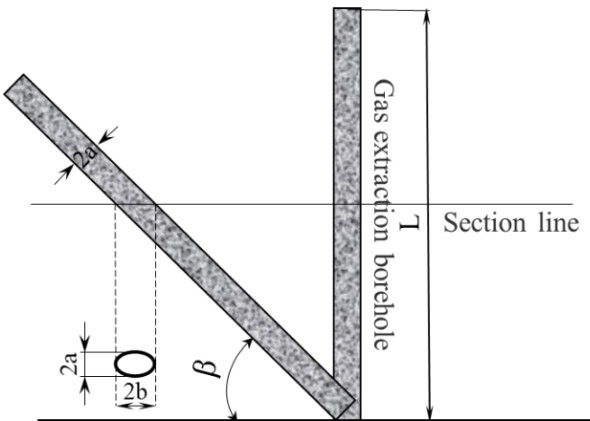

**Figure 2.** Mathematical analysis diagram of different drilling angle.

According to Figure 2, the relationship between the major semi-axis and minor semi-axis of ellipse is:

$$b = a / \sin \beta \tag{1}$$

where, $a$ is the length of minor semi-axis, $b$ is the length of major semi-axis, and $\beta$ is drilling angle.

Because it is difficult to obtain rock loose circle of bidirectional unequal pressure roadway, a perturbation solution with high accuracy is solved [17]. However, its solution is complicated, while the approximate solutions proposed by other methods [18–20] are based on the solution as the rock around the circular hole is completely elastic, lacking theoretical basis. Therefore, for elliptical drillings in engineering, the ellipse is treated as a circular section and analyzed using equations of circular drillings or directly by numerical simulation. In this paper, the complex variable function is used to analyze the area of rock loose circle around the elliptical drilling. In the elliptical area in the Z plane, the major axis is 2*a*, and the minor axis is 2*b*, as shown in Figure 3. The mapping function is [21]:

$$z = w(\zeta) = c\left(\zeta + \frac{m}{\zeta}\right) \tag{2}$$

where, $c = (b - a)/2$, $m = (b - a)/(a + b)$. Substitute $z = x + iy$ and $\zeta = \rho_\zeta e^{i\theta} = \rho_\zeta (\cos\theta + i\sin\theta)$ into Equation (2), the $\zeta$ plane parameter represents the functional expression of the ellipse in the Z plane.

$$\frac{x^2}{\left(cp\zeta + \frac{cm}{\rho\zeta^2}\right)^2} + \frac{y^2}{\left(cp - \frac{cm}{\rho\zeta^2}\right)^2} = 1 \tag{3}$$

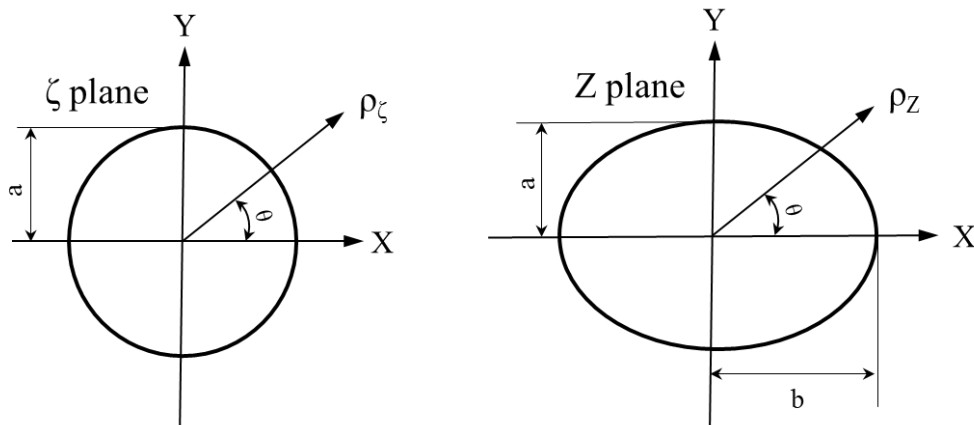

**Figure 3.** Mapping diagram of loose circle, (*a*) $\zeta$ plane, (*b*) $Z$ plane.

As shown in Figure 3, the radius of the loose circle at a certain angle in the $\zeta$ plane is equal to the radius of the coordinate point at the corresponding angle on the junction line of the elastic zone and plastic zone outside the circular orifice, so the radius of the loose circle outside the circular orifice in the $\zeta$ plane is [22]:

$$\rho_\zeta = R_0 \left\{ \frac{[p(1+\lambda) + 2C \cot\varphi](1 - \sin\varphi)}{2C \cos\varphi} \right\}^{\frac{1 - \sin\varphi}{2 \sin\varphi}} \times \left\{ 1 + \frac{p(1-\lambda)(1 - \sin\varphi)\cos 2\theta}{[p(1+\lambda) + 2C \cot\varphi]\sin\varphi} \right\} \tag{4}$$

where, $R_0$ is the radius of mapping plane, its value is 1 m, $\rho_\zeta$ is the corresponding polar coordinate polar radius of the junction line of the elastic zone and plastic zone in the $\zeta$ plane, $\theta$ is the polar angle, $P$ is the ground stress, $\lambda$ is the lateral pressure coefficient, $\varphi$ and $c$ are the friction angle and cohesion inside wall rock, respectively.

The elliptical function coordinates in the $Z$ plane are expressed with polar coordinates:

$$\begin{cases} x_z = \rho_z \cos\theta \\ y_z = \rho_z \sin\theta \end{cases} \tag{5}$$

From Equations (3)–(5), the relationship between the radius $\rho_z$ of loose circle in polar coordinates in the $Z$ plane and the radius $\rho_\zeta$ of loose circle in the $\zeta$ plane is:

$$\rho_z = c\rho_\zeta \sqrt{1 + \frac{m^2}{\rho_\zeta^4} + \frac{2m}{\rho_\zeta^2}\cos 2\theta} \tag{6}$$

The area of the loose circle is calculated based on the radius of the loose circle of the elliptical orifice.

$$S = S_1 - S_2 \tag{7}$$

where, $S_1$ is the area of the loose circle containing the elliptical orifice, according to the symmetry, $S_1 = 2 \times \int_0^{\frac{\pi}{2}} \rho_z^2(\theta) d\theta$, $S_2$ is the area of elliptical orifice, $S_2 = \pi ab$.

Substituting Equations (1), (4) and (6) into Equation (7), we get:

$$S = \frac{\pi}{2} a^2 \left(1 + \frac{1}{\sin\beta}\right)\left(A + \frac{1}{2}\right) + \left(\frac{1 - \sin\beta}{1 + \sin\beta}\right)^2 \int_0^{\frac{\pi}{2}} \frac{1}{(A + B \cos 2\theta)^2} d\theta \tag{8}$$

where,

$$\begin{cases} A = R_0 \left\{ \dfrac{[P(1+\lambda) + 2C \cot\varphi](1 - \sin\varphi)}{2C \cos\varphi} \right\}^{\frac{1 - \sin\varphi}{2 \sin\varphi}} \\[2ex] B = \dfrac{P(1-\lambda)(1 - \sin\varphi)}{[p(1+\lambda) + 2C \cot\varphi]\sin\varphi} \end{cases} \tag{9}$$

*2.2. Result Analysis*

According to Equation (8), the function curve with drilling angle as the variable and the area of the rock loose circle around drilling as the dependent variable is plotted, as shown in Figure 4. If the diameter of drilling remains unchanged, the area of the loose circle around the elliptical drilling decays negatively and exponentially with the increase in drilling angle. Therefore, the decrease in drilling angle is conductive to gas extraction. By comparing with the area of the rock loose circle around the circular borehole, the calculated value is quite different. The area of the coal loose circle around the circular borehole decreases linearly with the increase of inclination angle, and the attenuation range is small. Through the analysis of field measured data [23] and according to the calculation results of elliptical drilling, the area of the loose circle around coal is more reasonable.

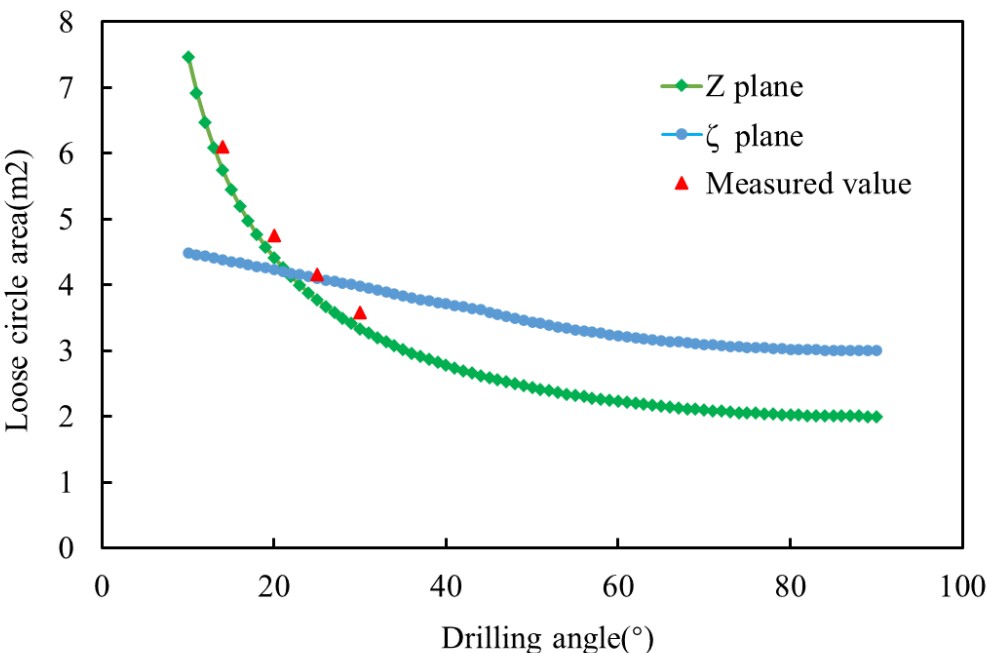

**Figure 4.** The change rule of loose circle area and drilling angle of the coal around the elliptic borehole.

## 3. Influence Pattern of Drilling Angle on Gas Extraction Effect

Yangmei Coal Mine 15# coal has a stable occurrence and simple structure, medium hard coal seam, the average thickness is 4 m, the average dip angle is 4°, it is a nearly horizontal coal seam, and the buried depth is about 500 m. The lithology of the roof and floor are siltstone. The strike longwall big height mining method is adopted, and the roof is managed by all caving methods. During mining, the average relative gas emission is 66.56 m³/t, and the average absolute gas emission is 4.16 m³/min. The gas pressure is in the range of 0.9~1.0 MPa. It belongs to an area with high gas pressure and high gas content.

In order to understand the influence pattern of drilling angle on the gas extraction effect, a geometric model is established using COMSOL Multiphysics, taking account of the coal seam conditions, mining technology, ground stress and design conditions for gas extraction in Yangmei 5 Mine 8406 working face, as shown in Figure 5. The following assumptions are made: (1) The coal body is isotropic. (2) The gas is ideal. (3) The gas-containing coal is saturated by a single phase. (4) The solid-gas coupling process of gas extraction is isothermal.

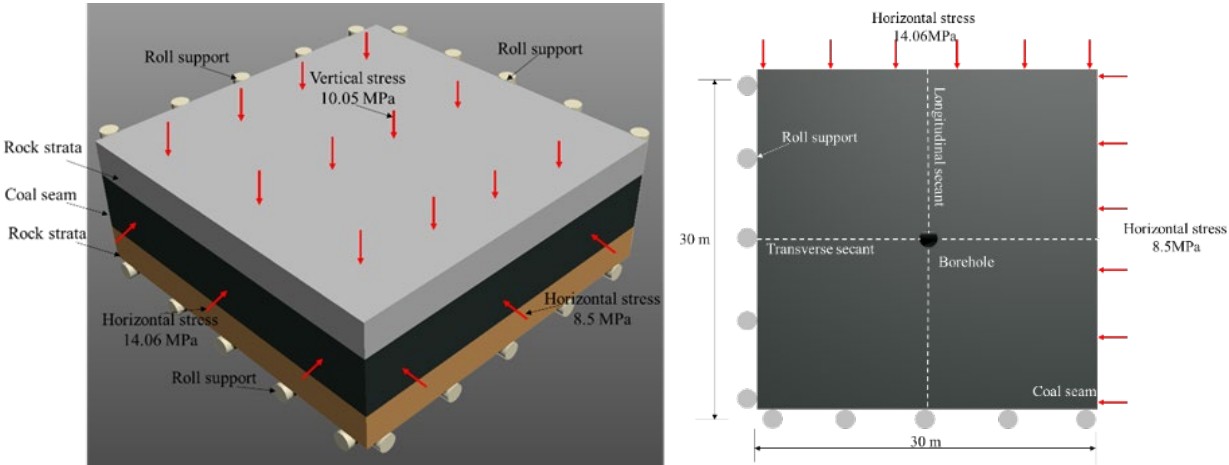

**Figure 5.** Numerical simulation model.

### 3.1. Equation of Dynamic Change of Coal Permeability

Neglecting the effect of free gas change on coal rock or porosity, we have the expression of coal permeability [16,19]:

$$k_g = \frac{k_\infty}{1+\varepsilon_v}\left[1+\frac{\varepsilon_v}{\varphi_0}\right]^3 \tag{10}$$

where, $\varphi_0$ is the initial porosity, $k_\infty$ is the initial permeability, and $\varepsilon_v$ is the volumetric strain of coal generated during the deformation process.

### 3.2. Determination of Fluid-Solid Coupling Equation

According to Darcy's law, mass conservation, the Klinkenberg effect and the Langmuir model, rock mechanics and elastoplasticity theory, the gas seepage Equation (11) and coal deformation control Equation (12) in the coal seam are constructed [24–28]:

$$\text{div}\left(-\frac{\beta p k_g}{\mu_g}\nabla p\right) + \beta\left[\frac{2\varphi p}{p_n} + \frac{2b_1\rho_c(1-A-B)p}{p+b_2} + \frac{(1-\varphi)p^2}{E_s p_n} - \frac{b_1\rho_c(1-A-B)p^2}{(p+b_2)^2}\right]\frac{\partial p}{\partial t} = Q_s - \frac{\beta_\alpha p^2}{p_n} \tag{11}$$

$$G\sum_{j=1}^{2}\frac{\partial^2 u_i}{\partial x_j^2} + \frac{G}{1-2v}\sum_{j=1}^{2}\frac{\partial^2 u_j}{\partial x_j \partial x_i} - \frac{(3\lambda-2G)}{3E_S}\frac{\partial P}{\partial x_i} + a\frac{\partial p}{\partial x_i} + F_i = 0 \tag{12}$$

where, $\beta$ is the compression factor of gas in coal seam in kg/(m³·Pa), $p$ is the gas pressure in Pa, $k_g$ is the effective permeability, $\mu_g$ is the dynamic viscosity coefficient of gas, $\varphi$ is the dynamic varying porosity of coal, $P_n$ is the atmospheric pressure, $\rho_c$ is the density of coal, $b_1$ is the ultimate adsorption capacity of coal in m³/kg, $b_2$ is the Langmuir pressure parameter of coal, $A$ and $B$ are the ash and moisture in coal, $E_s$ is the modulus of volume elasticity of skeleton particles of gas-containing coal in Pa, $Q_s$ is the mass of gas generated or absorbed per unit time in the unit volume of gas-containing coal in kg/(m³·s), $\alpha$ is the equivalent pore pressure coefficient of coal, and $\varepsilon_v$ is the volumetric strain of coal. Where $k_g = k_\infty(1 + b/p)$, $k_\infty$ is the initial permeability, $\varphi = 1 - (1 - \varphi_0)/(1 + \varepsilon_v)$, $\varphi_0$ is the initial porosity, $u_i$ is the displacement component of coal, $F_i$ is the force component applied to coal, and $G$ is the shear modulus of coal.

### 3.3. Model Establishment and Boundary Condition Setting

As shown in Figure 5, the length and height of the established model are both 30 m, the left boundary and the lower boundary conditions are set to roll support, the right and left boundary are the horizontal lateral ground stresses of 14.06 MPa and 8.5 MPa, and the top boundary is loaded with 10.05 MPa vertical ground stress. The middle borehole is the shape of an inclined borehole in a horizontal section, its major and minor axes vary with

the inclination angle, and the minor axis is constant $a = 0.11$ m, $b = a/\sin\beta$. The original gas pressure in the coal seam is 0.92 MPa, and the gas extraction negative pressure is $-10$ KPa. The parameters of the model of the coal seam are shown in Table 1. In order to facilitate data analysis, two secants are taken on the model.

**Table 1.** Model parameter.

| Parameter | Value |
|---|---|
| Shear modulus $G$/MPa | 90.3 |
| Poisson ratio $v$ | 0.16 |
| Density $\rho_c$/(kg·m$^{-3}$) | 1400 |
| Initial porosity $\varphi_0$ | 0.0456 |
| Gas dynamic viscosity coefficient $\mu_g$/(Pa·s) | $1.9 \times 10^{-6}$ |
| Equivalent pore pressure coefficient $\alpha$ | 0.1604 |
| Temperature $T$/°C | 20 |
| Atmospheric pressure $P_n$/MPa | 0.101 |
| Adsorption constant $b_1$/(m$^3$·kg$^{-1}$) | $36.492 \times 10^{-3}$ |
| Adsorption constant $b_2$/MPa | 1.48 |
| Ash content $A$ | 11.48% |
| Moisture content $B$ | 3.41% |
| Bulk modulus of coal skeleton Es/MPa | 300 |
| Initial permeability $k_\infty$/m$^2$ | $0.2748 \times 10^{-16}$ |
| Gas compression factor $\beta$/(k$_g$/(m$^3$·Pa)) | 0.987 |
| Gas adsorbed per unit volume of coal $Q_s$/kg/(m$^3$·s) | $30.98 \times 10^{-4}$ |

*3.4. Result Analysis*

3.4.1. Varying Pattern of Coal Permeability around Drilling with Inclination Angle

According to the model in Figure 5, the coal permeability around the elliptical drilling with a ratio of the major axis to minor axis 2 is calculated, and the coal permeability around drilling under the transverse and longitudinal secants through the center is plotted, as shown in Figure 6. At the same distance from the center of drilling, the coal permeability at the transverse secant is larger than that at the longitudinal secant, indicating that the volume strain of coal in the transverse position is larger than that in the longitudinal position, and fractures in coal grow, so the permeability of coal is improved.

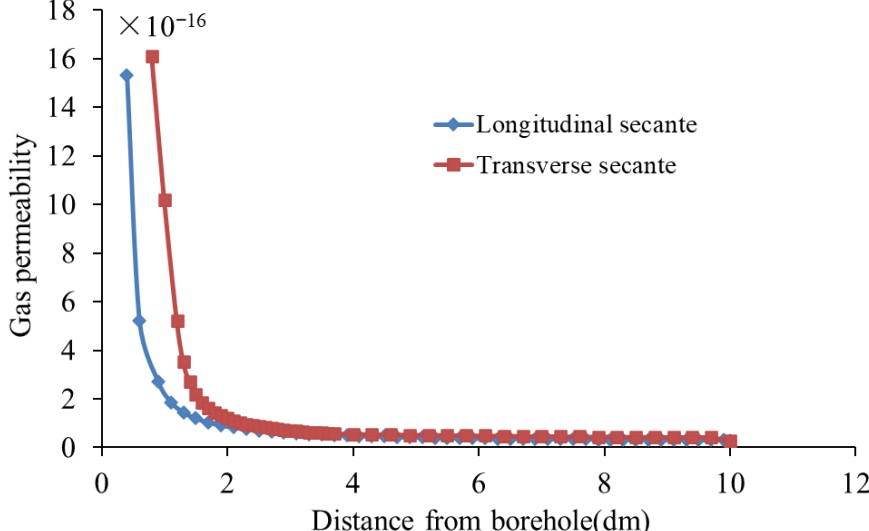

**Figure 6.** Regularity of permeability around borehole.

As shown in Figure 7, under longitudinal secant, within 10 dm from the center of drilling under the longitudinal secant, the permeability of coal around drilling increases with the reduction of the drilling angle. The coal permeability changes sharply at the orifice and is close

to the coal original permeability $0.027 \times 10^{-16}$ along the far away from the orifice, indicating that the volumetric strain of coal around drilling increases with the reduction of drilling angle, the increase of fractures in the coal improves the permeability of coal.

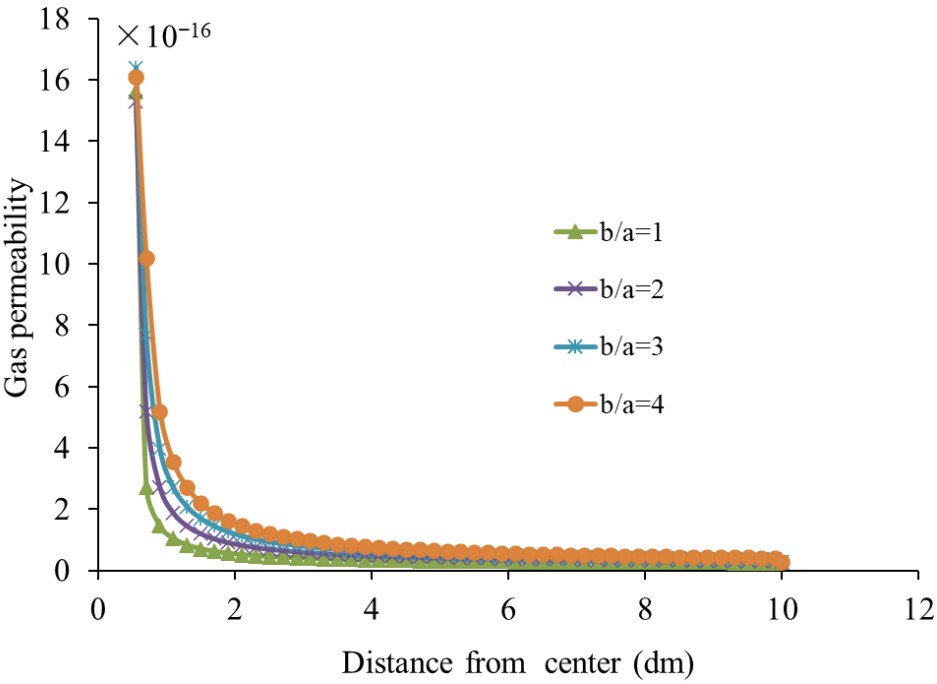

**Figure 7.** The permeability of coal with different drilling angles under the longitudinal secant.

3.4.2. Varying Pattern of Gas Pressure around Drilling with Inclination Angle

From the gas pressure distribution cloud atlas around drilling with different inclination angles as shown in Figure 8, the area of gas pressure reduction in coal around drilling increases as the inclination angle decreases.

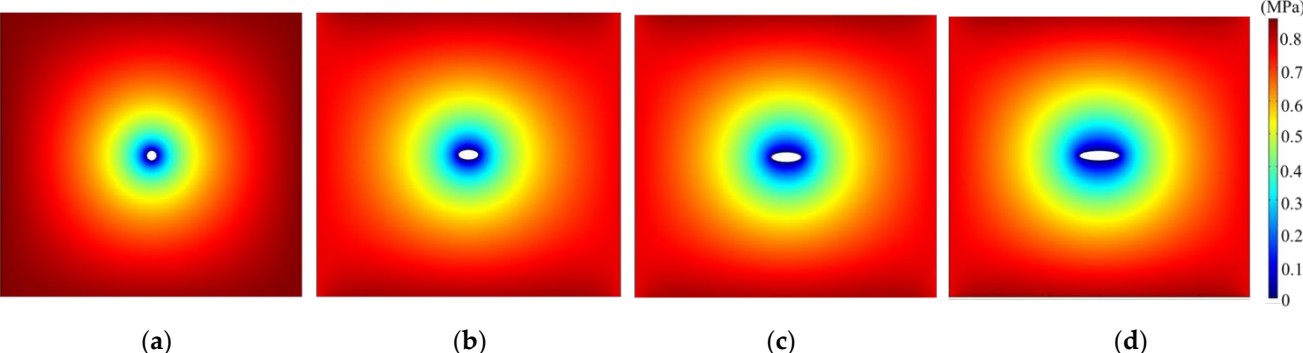

(**a**)　　　　　　　(**b**)　　　　　　　(**c**)　　　　　　　(**d**)

**Figure 8.** Gas pressure distribution of gas pressure in coal seam with different drilling angles. (**a**) $b/a = 1$, (**b**) $b/a = 2$, (**c**) $b/a = 3$, (**d**) $b/a = 4$.

The varying pattern of gas pressure in coal around drilling with different inclination angles under the transverse and longitudinal secants are shown in Figure 9, respectively. At the same distance from drilling, the gas pressure in coal decreases as the inclination angle reduces. At the same time, under the same gas pressure condition, the distance from the center of drilling increases as the inclination angle decreases. The above phenomena show that as the inclination angle decreases, the gas pressure in coal around drilling decreases, and the reduction area of gas pressure expands, which is conductive to gas extraction in coal seam.

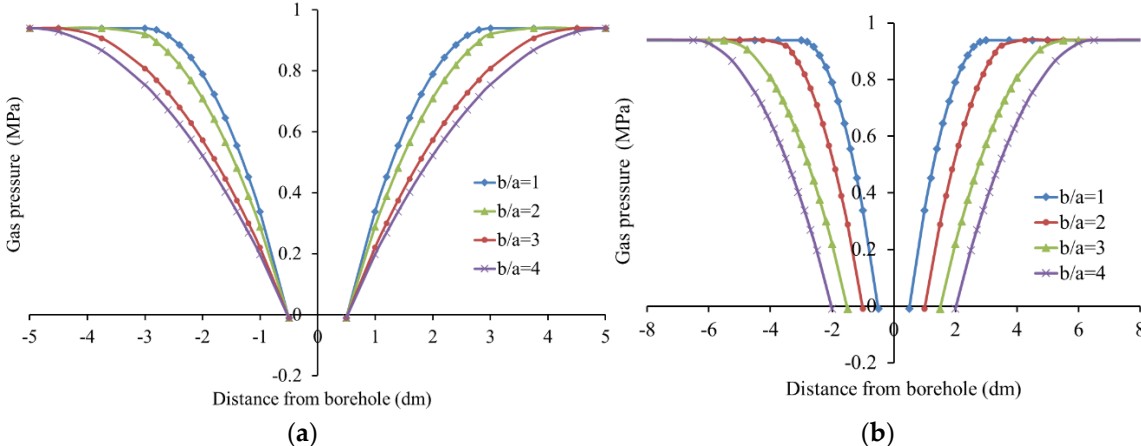

**Figure 9.** Gas pressure of coal around boreholes with different drilling angles. (**a**) longitudinal secant, (**b**) transverse secant.

## 4. Arrangement Optimization for BDR

From the above analyses, the smaller the drilling angle, the more favorable it is to the gas extraction in the coal seam. However, as the drilling angle decreases, it will cause stress concentration in coal around the elliptical drilling, which leads to the collapse of drilling and hinders gas extraction. Therefore, in order to facilitate gas extraction, there is an optimal drilling angle of the borehole; that is, there is a reasonable layer in the BDR that determines the inclination angle of the borehole.

According to reference [10], the tangential stress around the ellipse is:

$$\sigma_\theta = \lambda P_0 \frac{k^2(\sin\theta)^2 + 2k(\sin\theta)^2 - (\cos\theta)^2}{(\cos\theta)^2 + k^2(\sin\theta)^2} + P_0 \frac{(\cos\theta)^2 + 2k(\cos\theta)^2 + k(\sin\theta)^2}{(\cos\theta)^2 + k^2(\sin\theta)^2} \quad (13)$$

where, $\sigma_\theta$ is the tangential stress around the borehole, $\theta$ is the polar angle of the borehole, $P_0$ is the ground stress, $\lambda$ is the lateral pressure coefficient, $k$ is the ratio of the major axis to minor axis, and $k = b/a = 1/\sin\beta$ from Equation (1), $\beta$ is the drilling angle ($0° < \beta \le 90°$), $a$ is the minor axis of the ellipse, and $b$ is the major axis of the ellipse.

The ratio of the maximum horizontal primary stress to the minimum horizontal primary stress in mines in Shanxi is in the range of 1.5–2 [29]. From Equation (13), when the lateral pressure coefficient $\lambda = 2$, the tangential stress distribution around the elliptical drilling is shown in Figure 10:

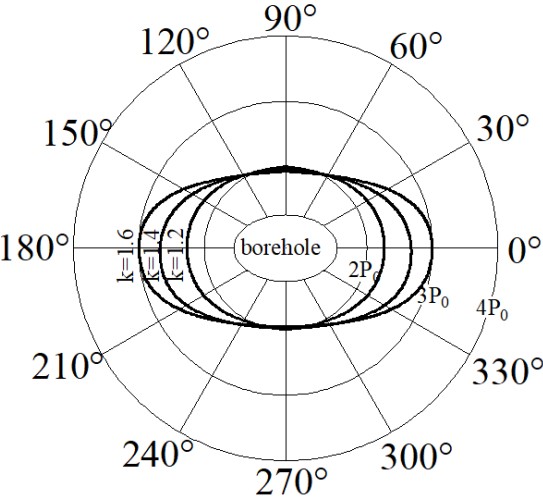

**Figure 10.** Tangential stress around elliptical borehole.

Let the maximum compression strength of rock be $\sigma_t$, as shown in Figure 10, and the tangential stress around drilling be the largest as the polar angle of the borehole at $0°$ and $180°$, which is the most prone to collapse. Therefore, only the tangential stress around drilling at $0°$ and $180°$ are considered, and Equation (13) is converted to:

$$\sigma_\theta = -\gamma P + P(1 + 2k) \tag{14}$$

In order to prevent the borehole from collapsing, the tangential stress around the borehole needs to be:

$$\sigma_\theta \leq \sigma_t \tag{15}$$

In summary, the optimal drilling angle that allows the maximum area of loose circle without collapse is:

$$\beta = \arccos\frac{2P}{\sigma_t - P + \lambda P} \tag{16}$$

As the layer in the BDR determines the drilling angle, as shown in Figure 1, the optimum distance between the BDR and the coal seam according to the influence of inclination angle of drilling on gas extraction effect is:

$$H = D\tan\beta - h = D\frac{2P}{\sqrt{(\sigma_t - P + \lambda P)^2 - 4P^2}} - h \tag{17}$$

where, $D$ is the unilateral extent of the coal seam for gas extraction in the BDR in m, $\beta$ is the drilling angle in $°$, $h$ is the thickness of coal seam in m, and $\sigma_t$ is the maximum compression strength of rock in MP.

## 5. Engineering Applications

Based on the above theoretical study, the field experiment was conducted in the Yangmei Coal Mine, where the average thickness of the coal seam was 4 m, the uniaxial compression strength of the coal seam was 14 MPa, the unilateral extraction range was 10 m. The horizontal ground stress was 11 MPa, and the lateral pressure coefficient was 2. The optimum drilling angle was calculated as $29°$; the optimal distance between the BDR and the coal seam was 10 m, which met the requirement of the maximum damage depth of the BDR at the mining face stipulated in the "Regulations on Prevention and Control of Coal and Gas Outburst". Moreover, it was compared with the gas extraction effect originally at 30 m between the BDR and the coal seam, as shown in Figure 11.

As shown in Figure 11, the average gas drainage volume of three groups of the gas drainage boreholes in the BDR has a spacing of 10 m and 30 m from the coal seam. The comparison of the average gas drainage volume of three groups of gas drainage boreholes in the BDR with the spacing of 10 m and 30 m from the coal seam is also shown. It can be seen that the gas drainage volume of the BDR with different spacing decreases with the increase of drainage time. In the BDR with the spacing of 10 m from the coal seam after optimization, the average gas drainage volume of group A, group B and group C boreholes are $0.0051\ \mathrm{m^3/min}$, $0.0040\ \mathrm{m^3/min}$ and $0.0068\ \mathrm{m^3/min}$, respectively. In the original BDR with the spacing of 30 m from the coal seam, the gas drainage volume of group A, group B and group C are $0.0021\ \mathrm{m^3/min}$, $0.0014\ \mathrm{m^3/min}$ and $0.0026\ \mathrm{m^3/min}$, respectively. The extraction rate of three groups of boreholes with the spacing of 10 m is about 60% higher than that with the spacing of 30 m. The above results show that the optimized BDR with the spacing of 10 m can significantly improve the gas drainage effect and further verify the reliability of the theoretical research.

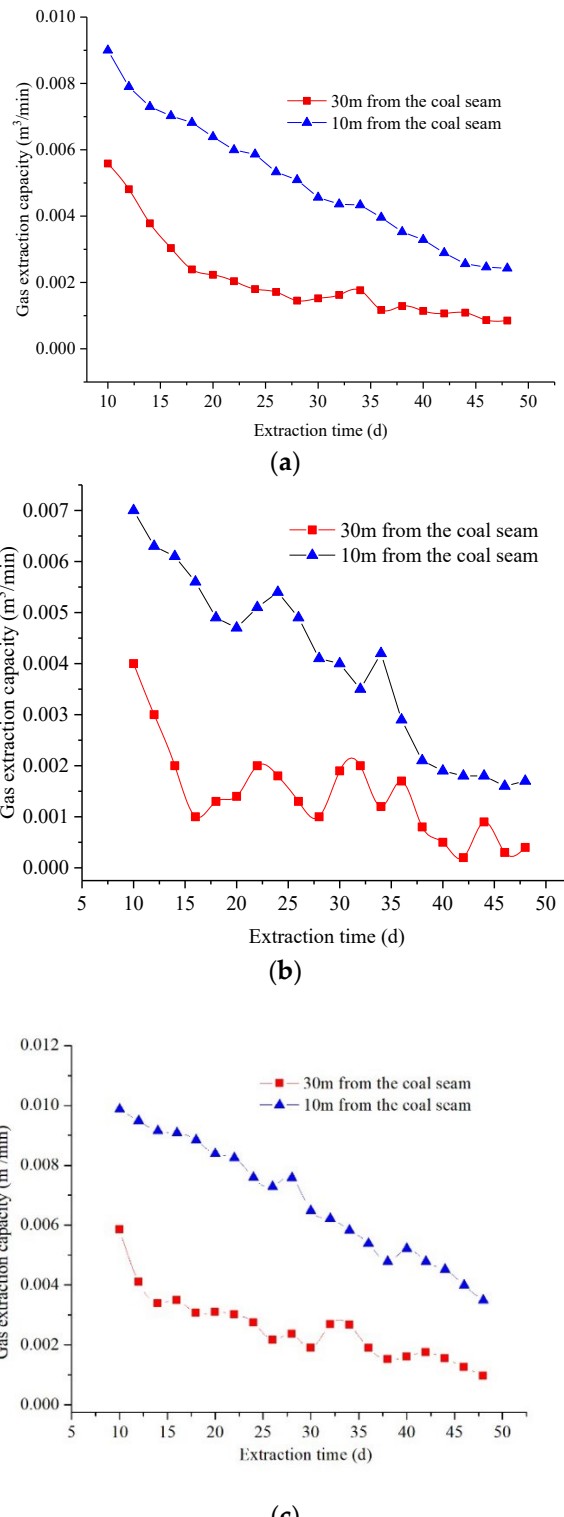

**Figure 11.** The gas extraction amount of lane with different distance to coal seam: (**a**) Boreholes of group A, (**b**) Boreholes of group B, (**c**) Boreholes of group C.

## 6. Conclusions and Discussions

Combined with the coal seam occurrence conditions are mining technical conditions, in-situ stress state and gas drilling drainage design of the Yangmei Coal Mine. This paper establishes a mathematical model and analyzes the influence of the change of the drilling angle on the area of the coal loose circle around the elliptical borehole. The fluid–solid

coupling model is constructed using COMSOL software. The model considers the dynamic response of coal permeability and volumetric strain with gas pressure, the Klinkenberg effect of gas is considered, and the effect of the change of elliptical drilling angle on the pressure relief effect of the coal seam is studied. Finally, the reasonable position of the BDR is determined by measuring the in-situ stress. It can provide theoretical guidance to the site.

(1) The complex function method is used to analyze the area of the loose circle around the elliptical borehole, which is similar to the field measured results, and is quite different from the calculated value of the area of the loose circle around the circular borehole;

(2) With the increase in drilling angle, the area of the loose circle and permeability of coal around the borehole decays negatively and exponentially. It shows that with the decrease in drilling angle, the volumetric strain of coal around the borehole increases, the cracks in the coal increase, and the permeability of coal is improved. At the same time, with the decrease in drilling angle, the gas pressure of coal around the borehole decreases, and the area of gas pressure reduction expands, which is conducive to the drainage of gas in the coal seam;

(3) Through theoretical calculation, it is found that the coal gas pre-drainage effect is better when the drilling angle of the 8406 longwall face of Yangmei Coal Mine is $29°$, and the distance between the BDR and the coal seam is 10 m. Through field measurement, the maximum average single hole gas drainage volume is 0.0068 $m^3$/min; the extraction rate of the boreholes with the spacing of 10 m is about 60% higher than those with the spacing of 30 m.

The fluid–solid coupling mathematical model of coalbed methane migration needs to be further improved. Coal is a complex porous medium, which itself has cleats and end cleats. The model established in this paper does not consider the influence of the primary fractures in the coal seam and the secondary fractures after gas release. This paper only aims at the variation characteristics of extraction boreholes under the condition of fixed coal seam dip angle, but in practice, the dip angle of coal seam changes. Follow-up research needs to supplement the variation characteristics around extraction boreholes under different dip angles of the coal seam.

**Author Contributions:** Conceptualization, Y.Y. and P.H.; methodology, Y.Y.; software, Y.Y.; validation, Y.Y., P.H. and Z.Z.; formal analysis, Y.Y.; investigation, Y.Y.; resources, Y.Y.; data curation, P.H.; writing—original draft preparation, Y.Y.; writing—review and editing, Y.Y.; visualization, W.C.; supervision, P.H.; project administration, Y.Y.; funding acquisition, Y.Y. All authors have read and agreed to the published version of the manuscript.

**Funding:** This research was funded by the Natural Science Foundation of Shanxi Province, China grant number 201903D121070, 201901D111305, 20210302123336.

**Informed Consent Statement:** Written informed consent has been obtained from the patient(s) to publish this paper.

**Data Availability Statement:** All data, models, or codes generated of used during the study are available from the corresponding author by request.

**Conflicts of Interest:** All the authors of this manuscript have approved the article's submission for publication, and there are no conflicts of interest to declare. This paper has not been published elsewhere and is not under consideration by another journal. The authors declare no conflict of interest.

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
