# Peer review of "The Method of Determining Layer in Bottom Drainage Roadway Taking Account of the Influence of Drilling Angle on Gas Extraction Effect"

_sustainability, doi:10.3390/su14095449_

Round 1

Reviewer 1 Report

I thought this manuscript was excellent and it was favorably reviewed.

This paper discussed the effect of drilling angles on gas extraction by numerical simulation, and combined with the measured data, the layer in floor gas extraction roadway was determined. Overall, the research was conducted scientifically but the paper is not well written.

-The whole paper needs to improve the presentation and the English writing.

-The effect of surrounding rock pressure on hole is very significant, why did the author not consider this issue?

-Some assumptions and theories are only applicable to specific situations, so the conclusions are not universal, which should be explained in the paper.

- The content of experiment and measured data should be fully supplemented and discussed.

Reviewer 2 Report

The assumption of the drilling projection given by the authors in line 87 is a great simplification and can only be made if the conditions of parallel position of the BDR gallery to the coal seam are met in any other case it is different, it is a great simplification. In reality, the seam may be running sub-fall with respect to the sidewalk and this fall may change as a function of distance, not to mention cases of cracks and faults and wedging or dismemberment of the seam (assuming homogeneity). In all these assumptions it will be more complicated. The authors should show that for a proper analysis, all non-trivial cases of seam position relative to the sidewalk have deviations of negligible or relatively small values otherwise the entire proposed text is basically a single case study. 
The entire text below is based on this assumption therefore the proposed calculations likewise reflect this case. 
Later in the text (lines 144) we learn that by assumption the deck is homogeneous, filled with one gas phase, etc. This further narrows the application of the method discussed by the authors. 
In fact, in subsection 2.2 one can guess that as the cross-section of the borehole increases (at an angle the projection shows a more and more elongated ellipsoid) it will obviously have an impact on the possible interaction with the deposit, for this one does not have to harness mathematics.
Line 239 - the hole is stressed not only by the deposit but also by the surrounding rocks, which in the case of deposits surrounded by e.g. marl rocks may be of great significance for the hole behaviour and its durability. The authors do not seem to have considered this at all.
As a matter of fact, while reading the text I was hoping for more data in chapter 5, because it may help us verify the data, but... here is only one small graph and a few sentences.
And in the conclusions it turns out that it is not the angle of inclination but the closeness of the sidewalk to the deck that matters most. 
And at the end I will add that References has few publications and is badly formatted.
The text seems to me to be written chaotically, in fact behind the facade of mathematical formulae there are simple assumptions and it is difficult to estimate what the authors mean, especially that the conclusions do not leave any illusions (only 3 points, not even further research was indicated).

Reviewer 3 Report

SUMMARY

The paper deals with coal seam gas pre-pumping through BDR wells, in order to prevent and control the explosion of coal and gas. In this article, the mechanism of influence of drilling angles on the gas extraction effect has been analyzed by a theoretical model of rock loose circle area around elliptical drilling, using COMSOL to carry out the numerical calculations, taking into consideration important factors such as permeability and gas pressure.

FINDINGS

-The results showed that the effective extraction length increases with the decrease in the distance between the BDR layer and the coal seam prior to pumping.

-The increase in the drilling angle caused a significant decrease in the area and permeability of the loose circle.

-A decrease in the perforation angle caused an increase in the stress in the main axis of the ellipse, leaving the perforation closer to collapse.

-An optimized method is proposed to determine the layer in the gas extraction roadway, combining the soil stress measures with the effect of multiple other factors.

STRENGTHS

Good organization of the article.

Good theoretical explanation and presentation of the model.

WEAKNESSES

The paper is well organized and correctly addresses the subject of study. But some minor changes would be necessary for the acceptance of the manuscript. Thus, presentation of the article could be improved:

-Some formulas do not appear correctly centered in the manuscript.

-The introduction section, but especially the conclusion section, seem too summarized. Some discussion is missing in the conclusions.

-Figure captions are sometimes read "Fig." and other "Figures". Same for table 1, which reads "Tab." and shows a bit rare.

-Improve the formatting of figure 4. It could show axes and text in black, instead of gray.

-Improve the formatting of figure 5. Different font sizes, some difficult to read.

Round 2

Reviewer 2 Report

Dear Authors, I see many contributed changes to the proposed manuscript. I still believe that the assumptions of the text limit the universality of the proposed solutions, but I have no further comments .